# Chronic Pain in Spanish Wildland Firefighters

**DOI:** 10.3390/jcm11040989

**Published:** 2022-02-14

**Authors:** Fabio García-Heras, Jorge Gutiérrez-Arroyo, Patxi León-Guereño, Belén Carballo-Leyenda, Jose A. Rodríguez-Marroyo

**Affiliations:** 1VALFIS Research Group, Department of Physical Education and Sports, Institute of Biomedicine (IBIOMED), University of León, 24071 León, Spain; fgarh@unileon.es (F.G.-H.); abcarl@unileon.es (B.C.-L.); 2Health Pass Research Group, Department of Physical Activity and Sports, Faculty of Psychology and Education, University of Deusto, 20012 Donostia, Spain; patxi.leon@deusto.es

**Keywords:** chronic pain, wildland firefighters, physical employment, physical exercise, occupational health

## Abstract

The work performed by wildland firefighters (WFFs) is very demanding owing to the conditions in which they have to operate. It has been reported that these professionals walk long distances over unstable and steep terrain carrying heavy loads, handle tools manually and repeatedly and are subject to a high level of thermal stress. Under such conditions, the risk of developing chronic pain (CP) is high, although despite this, there are no available data pertaining to CP among WFFs, to the best of our knowledge. As such, the aim of this study is to describe CP in Spanish helitack crews, for which purpose 221 WFFs (203 men and 18 women) completed an online self-report questionnaire. Approximately 60% of WFFs reported suffering from CP, of which 45.5% had CP in more than one body region at the same time. Age and length of service were associated with the probability of suffering from CP. Likewise, the age and height of WFFs and weight of the protective equipment used increased the prevalence of CP. Lastly, gender and job position affected CP location, with women and forepersons reporting greater prevalence of CP in the lower limbs. To conclude, the results obtained suggest the major prevalence of CP among Spanish WFFs, with current data suggesting in turn the importance that age, stature, gender, length of service, weight of protective equipment and job position have on the prevalence and location of CP.

## 1. Introduction

The tasks performed by first responders (e.g., firefighters, police, soldiers) are eminently physical and tend to be conditioned by the changing situations and uncertainty caused by emergencies [1]. This is the case with the work carried out by wildland firefighters (WFFs), who habitually walk over long distances carrying heavy equipment up to 20–25 kg in weight [2] systematically hansdle tools during long working days that may exceed 10 h [3] and have to deal with a high level of thermal stress owing to the environmental conditions in which they are deployed [4]. Such circumstances mean that the physical demands of the work being undertaken by these professionals are high [4], which might increase the risk of becoming injured or of an occupational hazard in the short term [5], while there might be more probability of suffering from chronic pain and invalidity in the long term [5].

Many studies have described the high probability of first responders becoming injured while performing their work [6,7,8,9,10,11]. It has been reported that one firefighter is injured every 8 min while on active service [11] and that approximately 20% of firefighters may become injured during their working day [12]. Moreover, it has been informed that the body regions that are most frequently injured are knees (14%), lumbar region (13.5%) and shoulders (8.5%) [12]. To the best of our knowledge, there has only been one study that has analysed injuries among a large group of WFFs [10]. In this study, a total of 1304 injuries were analysed over a 5-year period, and it was noted that the main causes of injury were falls (28%) and the use and carrying of heavy tools (22%). The body regions most affected were lower limbs (35%), upper limbs (22.5%), neck (18.7%) and back (9.3%) [10].

Injuries at work may be one of the reasons why workers report the onset of pain or persistent discomfort over long periods of time (≥3 months) [13,14]. This chronic pain (CP) may end up affecting workers’ quality of life and work performance [14], above all in the more physically demanding professions [15]. Thus, it has been reported that 33–40% of American firefighters have experienced some type of chronic back pain that has conditioned their professional work [8,16]. Although the use of heavy loads, performance of repetitive tasks and carrying of tools above the head has been linked to the probability of suffering from CP in male forestry workers [17], to the best of our knowledge, CP has not been studied in WFFs. The analysis of its prevalence among these workers might highlight the need to perform interventions aimed at preventing this condition. In this sense, the beneficial role of rehabilitation [18], physical exercise [19,20] and nutrition [21] on CP management has been evidenced. Therefore, the aim of this study is to analyse CP in Spanish WFFs and explore its potential association using demographic and occupational parameters. It was hypothesized that the prevalence of CP in WFFs would be high due to the work’s characteristics performed by these professionals. In addition, we believe that variables, such as subjects’ age, sex or experience, would affect the probability of suffering CP or its body distribution.

## 2. Methods

This was a retrospective, cross-sectional study based on an online self-report questionnaire. The questionnaire was drawn up by specialists in occupational health at the University of León and the University of Deusto (Spain) on the basis of questionnaires previously used in the literature [11,22,23]. Moreover, forest fires experts were consulted in order to contextualise the work carried out by WFFs and concentrate on the questions to be asked. A questionnaire was then put together containing 45 questions about occupational injuries (details not included in the manuscript) and CP suffered by WFFs (see Appendix A), and comprised four sections: (i) demographic features, (ii) features linked to physical activity undertaken by WFFs, (iii) occupational injuries and (iv) chronic pain. The first section contained information about gender, age, weight, height, job position, tools used at work and length of service as a WFF. The second section focused on ascertaining the degree of physical activity undertaken by those interviewed. The purpose of the third section was to gather information about the type and location of occupational injuries suffered by WFFs. The questions contained in the last section were designed to try to identify the prevalence and location of CP. CP was defined as persistent pain greater than 3 months duration that might have been the result of disease, injury, trauma, surgery or unknown origin [13]. Likewise, the concept of pain was recently recommended by the International Association for the Study of Pain (IASP) [24]. Direct closed-ended questions (yes, no or do not know) or open-ended questions were used, while multiple-choice questions were also used where subjects marked different options separately. The time required to complete the questionnaire was no longer than 15 min.

Data were collected over February and March 2021, and the questionnaire was distributed in Google Forms via email lists of WFFs. Specifically, it was sent to all members (*n* = 557) of the Spanish Forest Fire Reinforcement Brigades (BRIF), who are elite WFFs transported by helicopter to wildfires, these helitack crews being located in different work areas within Spain. Apart from the questionnaire, WFFs received instructions about how to complete it and were also offered assistance in dealing with any doubt that might emerge while doing so. They were also provided with information about the purpose of the study, which was approved by the Ethics Committee of the University of León (Spain) (025-2020, 22 July 2020). 

Once the data had been collected, the resulting database was then reviewed so as to deal with any possible inconsistencies, duplicate information or dubious identifications. The results obtained were analysed by taking into account gender (male or female), age 18–35, 36–45 and >45 years), experience (≤10 and >10 years) and subjects’ job position (specialist, foreperson or crew leader). Moreover, bearing in mind the positive effect of physical activity or specific training on injury prevention [15,25,26], results were also analysed by taking into account the number of hours of physical activity set aside a week, and whether subjects carried out some specific training to prevent injury (yes/no). Physical activity was considered any exercise/training performed by WFFs in their leisure time. Taking into account the previous recommendations established for healthy adults [27], physical activity was classified as low (≤3 h), moderate (3–7 h) and high (>7 h). Lastly, the body regions where CP was reported were grouped into the following categories [8,28]: (i) lower limb (LL), which includes hip, knee and ankle; (ii) upper limb (UL), including neck, shoulder, elbow, hand and wrist; and (iii) back, including lower back and thoracic spine.

### Data Analysis

Demographic quantitative data are displayed as mean ± standard deviation, and the prevalence of categorical data responses is reported in terms of frequencies and percentages. A Mann–Whitney *U*-test and Kruskal–Wallis *H*-test for one-way analysis of variance were used to determine significant differences between group responses regarding quantitative data. If any differences were present, a post hoc pairwise test for multiple comparisons was used to confirm significance. Pearson’s Chi-square analysis was performed to assess whether the distribution of categorical variables differed from one to another. The Cochran *Q* test was used to compare the frequencies between body regions with chronic pain (i.e., LL, UL and back) within each categorical variable, while standardised residuals were assessed for significant differences between group responses and expected counts where significant associations were identified. Subsequent odds ratios and 95% confidence intervals (CI) were calculated to analyse associations, and the significance level was set at *p* < 0.05. Lastly, statistical analyses were performed using SPSS (v25, IBM Corporation, Armonk, NY, USA).

## 3. Results

In total, 221 WFFs (18 women and 203 men) took part in the study, which represented 40% of potential subjects who might have completed the questionnaire. All the women who were sent the questionnaire completed it, although only 38% of men did so. 

Of the total number of interviewees, 59.7% reported suffering from CP (Table 1). Significant differences were found in terms of age (U = 3554, *p* < 0.001) and experience (U = 3899, *p* < 0.001) between WFFs who reported CP and those without CP. Male WFFs with CP tended to be older (U = 3124.5; *p* < 0.001) and with higher work experience (U = 3329.5, *p* < 0.001) than those without CP. However, there were only significant differences (U = 13.0, *p* = 0.020) in the age for females WFFs (Table 1).

Of the 132 interviewees who stated having suffered from CP, 54.5% reported pain in a single body region, whereas 45.5% reported suffering from CP in more than one body region at the same time. Age (37.2 ± 5.6 vs. 38.7 ± 6.5 years), weight (78.5 ± 11.6 vs. 76.9 ± 10.5 kg) and work experience (11.1 ± 5.6 vs. 11.2 ± 5.2 years) were similar between both groups. However, WFFs with CP in more than one body region were (U = 2520.0, *p* = 0.022) shorter (174.9 ± 6.5 vs. 177.6 ± 7.0 cm) and used protective equipment that weighed significantly (U = 1564.5, *p* = 0.006) more than that used by WFFs with CP in just one body region (12.7 ± 5.4 vs. 10.1 ± 4.2 kg).

The prevalence of CP in WFFs according to gender, job position, age, physical and preventive training and experience is shown in Table 2. There was a significant association (χ^2^ = 20.727, *p* < 0.001) between CP and the age ranges analysed (i.e., 18–35, 35–45 and >45 years). The prevalence of CP in a body region was greater than that expected in age groups from 36–45 (40%) and >45 years (50%), while WFFs over 35 years were four times more likely to suffer CP in a body region (OR = 4.03, 95% CI [2.04–7.94]). An association was also found between CP and work experience (χ^2^ = 7.746, *p* = 0.021), insofar as the probability of suffering CP in a body region was greater in cases where WFFs had over 10 years experience (36.7% vs. 26.9%). These WFFs faced double the risk of suffering CP over those with less than 10 years experience (OR = 2.22, 95% CI [1.28–3.87]). No significant associations were found between CP and gender (χ^2^ = 0.007, *p* = 0.933), level of physical activity (χ^2^ = 3.305, *p* = 0.508) or performing preventive training (χ^2^ = 1.666, *p* = 0.435).

Overall, distribution of CP across the different body regions proved to be similar (Table 3), although female WFFs evidenced greater prevalence of CP in the LL (Q = 9.250, *p* = 0.010). Furthermore, an association between the area of pain and job position was found (χ^2^ = 6.005, *p* = 0.046), insofar as a larger percentage of forepersons reported suffering from CP in the LL than expected. The >45 year group reported CP mainly in the UL (χ^2^ = 14.600, *p* < 0.001, whereas in the 36–45 year group this was more prevalent in LL (χ^2^ = 6.811, *p* = 0.033) and back (χ^2^ = 1412.136, *p* = 0.002). Those interviewed with >10 years experience reported having had CP mainly in the UL. Lastly, a significant association between the probability of suffering from CP in more than one body region and the age of WFFs was noted (χ^2^ = 30.438, *p* = 0.007). The percentage of WFFs noted with CP in the UL and back or in both regions and LL was higher than the expected count in WFFs >45 and >35 years, respectively (Figure 1).

## 4. Discussion

The main finding of the present study was to determine the high prevalence of CP among Spanish WFFs analysed and the negative effect that length of service and age has on the probability of their suffering from CP. The prevalence observed (~60%) seem to be unaffected by the gender of WFFs, and in fact was very similar to what had previously been reported (~57%) in firefighters [8] and higher than that noted among other safety personnel (e.g., paramedics, polices, dispatchers) (~40%) [29]. One might think that the greater physical demands of the tasks performed by WFFs [1,2,3,30] would lead to a greater probability of CP [8], although Carleton et al. [29] found a higher prevalence in paramedics and correctional officers (~45%) than those findings in firefighters (~35%). This highlights the importance that other factors, such as age [16,31,32], length of service [31], work stress [16,29], physical fitness and subjects’ anthropometrics features [16], might have on the appearance of CP. 

The length of service of WFFs (i.e., greater work exposure time) was associated with CP, with our data pointing to the existence of a threshold beyond which WFFs might become more prone to suffering from CP. As such, WFFs who reported having suffering from CP had over 10 years of work experience (Table 1), doubling the risk of having it over those with <10 years. Systematic repetition of the specific movements used by WFFs (e.g., use of manual tools and carrying heavy loads) in the course of their working day [30] might involve repetitive injuries on to the musculoskeletal system resulting from overuse [31] and increase the probability of suffering from CP [17]. In this regard, Negm et al. [31] showed how firefighters with over 15 years’ experience in the firefighting service evidenced a higher incidence of musculoskeletal disorders.

Our results also highlight the importance of WFFs’ age in terms of the prevalence of CP, confirming previous findings obtained in firefighters [16,31,32]. In the present study, WFFs who did not report having suffered from CP were of an average age below 35 years, whereas those who did report having had CP were closer to 40 years (Table 1). These subjects had a greater probability (~4 times greater) of suffering from CP than their younger colleagues, and this would be located in different body regions at the same time (Figure 1). This may be down to the fact that older subjects become injured more frequently [10,17]. Specifically, similar features to those studied here have been analysed in American WFFs, insofar as a larger number of injuries was also noted in those subjects >33 years [10], and it has also recently been noted that firefighters >42 years are more prone to musculoskeletal disorders and suffer more severe LL and back disability than firefighters <42 years [31]. Moreover, previous studies [32] have found a positive link between firefighters’ age and injury duration, pointing out that older firefighters would need longer to recover. This study established the fact that an increase in age by one year in subjects might entail a 7% increase in injury duration [32]. Conversely, one might speculate that the increase in age of the WFFs who took part in this study could entail a reduction in their physical fitness, which in turn might increase the demands of the task they habitually perform and thus mean greater risk of injury and of suffering from CP [33]. In this regard, a reduction in aerobic capacity and the specific performance of firefighters as their age increases was analysed [34,35]. As such, a certain limitation on the part of subjects in the performance of their work might be expected, and in backing up this idea, it has recently been pointed out that firefighters >45 years versus those <45 years evidence more physical limitations during their working day [8]. Taken as a whole, these results would highlight the need to put injury-prevention programmes into place for older WFFs [15,25,36].

Approximately 33% and 27% of the WFFs in this study had CP in one or more body regions, respectively (Table 2). The probability of suffering from CP in different body regions was associated with age and height of WFFs and the weight of the protective equipment used. These variables might have contributed towards an increase in the demands of work carried out by them. Typically, these professionals tend to carry equipment during long working days that may weigh between 10–25 kg [2,30], meaning that smaller subjects might well require more energy per body mass unit to carry the load [37], and this may have contributed to the fact that the latter would become more tired and predisposed to getting injured [10,17]. Conversely, the weight of the protective equipment used by WFFs who had reported suffering from CP in more than one body region was ~2.5 kg heavier than that of subjects who had reported CP in just one body region. Although this difference might seem small, it represents ~10% of the maximum weight carried by WFFs when they are deployed [2,30], and might have increased the amount of energy used to undertake the activities carried out by 25–30% [12,38]. Thus, it makes sense to suggest that these WFFs might have experienced greater overload and fatigue in the course of work, which may have given rise to a large number of musculoskeletal complaints [16] and overuse injuries [31], in turn increasing the chance of suffering from CP in more than one body region. 

CP distribution across the different body regions was uniform (Table 3). These results may possibly have been conditioned by the habitual location of injuries in WFFs [10], although the CP distribution of greater prevalence of CP in LL vs. the back in women was found when analysing CP distribution according to gender (50% vs. 15%). A similar prevalence of LL injuries (~50%) has been reported in American female firefighters [39] and soldiers [40]. The anthropometric features of women (e.g., hip rotation, high knee valgus and more fat tissue) [1,25] might give rise to some potential biomechanical differences with men and condition a large number of LL injuries [41]. Thus, studies conducted on soldiers have identified greater prevalence of LL injuries in women, and more back injuries in men [1,40]. On the other hand, all WFFs in Spain have to carry the same load when they are deployed—nothing is adapted especially for women. This, combined with the fact that women have less muscle mass and tend to be shorter (~8 cm shorter in the case of the female WFFs in this study) [1] may have increased the demands of the work carried out [37], making musculoskeletal LL injuries possible [1,39,40] and, as a result, a greater probability of suffering from CP in this body region. Thus, literature on the subject often recommends undergoing specific physical training programmes to reduce the number of injuries and increase the physical fitness of female first responders [25,42,43,44]. 

CP distribution according to WFF job position also differed. Homogeneous distribution was noted in the case of WFF specialists, although CP in the LL was found to be more common in forepersons and crew leaders (Table 2). Previous studies have tended to link injury location with the types of task performed by workers [10,29], and in this regard, WFF specialists have to walk over long distances carrying heavy loads, which may lead to an increase in LL injuries [10,40]. In addition, these subjects work using hand tools, performing repetitive movements above shoulder level, with the trunk flexed forward, and also performing twisting movements of the trunk, which increases the chance of becoming injured both in the UL and back [17,40]. Under such conditions, it is to be expected that WFF specialists would result in uniform CP distribution across the different body regions. However, conversely, both forepersons and crew leaders mainly undertake organisational, monitoring and control work of operations, covering long distances while carrying heavy loads over uneven, unstable and steep terrain, which increases the risk of sprain or strain [10]. Moreover, these subjects tend to be older and have more work experience than specialists, which might have increased the chance of suffering from more severe LL disability [31], increasing the prevalence of CP in the LL.

Physical activity carried out by WFFs was not associated with the incidence of CP (Table 3), despite the fact that there is evidence to suggest that an increase in such activity may be beneficial in reducing the impact of chronic musculoskeletal pain [19,20]. This may possibly be due to the fact that all the WFFs who took part in the study had undertaken a physical training programme as part of their working routine, which might have limited the potential beneficial effect of physical activity. Future studies should include an in-depth analysis to determine the benefits of the type and amount of training undertaken in terms of the prevalence of CP. Neither was any association found between preventive training and CP. Having said this, it was noted that a large number of WFFs who reported suffering from CP did undertake this type of training, possibly to reduce their musculoskeletal pain [20].

Lastly, the present study is not without its limitations. First, this was a retrospective self-report survey, and it is possible that recall bias was present. In addition, CP was self-diagnosed by the WFFs; although they were instructed to precisely define CP, this fact might affect the results obtained. Second, the questionnaire did not detect either the causes that might have provoked the CP or CP characteristics. Therefore, further studies should analyse these aspects. Finally, another potential limitation of the study was the sample size. Although 203 male WFFs took part, they only represented 38% of the potential subjects who might have completed the questionnaire, and so an increase in sample size might have affected the results obtained. Conversely, although only 18 female WFFs took part, these made up all the female members of the Spanish Forest Fire Reinforcement Brigades and, on the other hand, the data provided in this study were limited to informing about CP suffered by helitack crews. We had no access to information pertaining to other brigades who work in forest firefighting throughout Spain. Despite these limitations and due to rare existing literature in this area, the results obtained in the present study may constitute an accurate approximation in describing CP in Spanish helitack crews, which might have potential applications in occupational medicine.

## 5. Conclusions

In conclusion, this is the first study to have analysed CP in Spanish WFFs. The results obtained demonstrate the high prevalence of CP among WFFs (~60%), with similar results being found in female and male WFFs, although gender and job position were variables that affected the location of CP. As such, female WFFs and forepersons evidenced CP more commonly in the LL, while conversely, both age and length of service increased the chance of suffering from CP. WFFs >35 years were four times as likely to suffer from CP, and those who had worked >10 years in wildland firefighting faced twice the risk of suffering from CP. Lastly, an association was found between the weight of the protective equipment used, age and height of WFFs, with the chance of suffering from CP in different body regions at the same time. These findings might provide relevant information to assist in the design of specific training programmes to prevent and improve CP in WFFs.

## Figures and Tables

**Figure 1 jcm-11-00989-f001:**
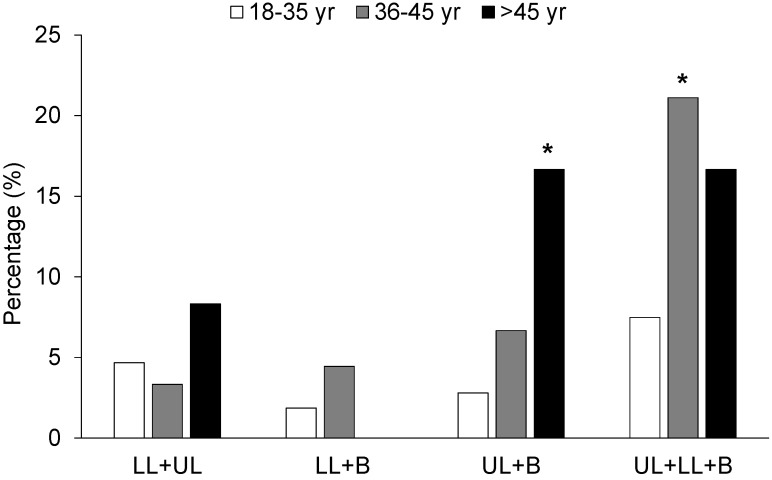
Distribution of chronic pain in more than one body region according to age range of wildland firefighters. LL: lower limb; UL: upper limb; B: back. *, higher count in this category than that expected (*p* < 0.05).

**Table 1 jcm-11-00989-t001:** Sociodemographic variables according to chronic pain and gender (mean ± SD).

	Total		Female		Male	
	NoCP (*n* = 89)	CP (*n* = 132)	NoCP (*n* = 7)	CP (*n* = 11)	NoCP (*n* = 82)	CP (*n* = 121)
Age (yr)	34.1 ± 6.3	38.0 ± 6.2 *	30.6 ± 5.8	40.4 ± 8.3 *	34.4 ± 6.2	37.8 ± 6.0 *
Height (cm)	177.6 ± 7.0	175.6 ± 6.1	169.9 ± 6.4	168.7 ± 6.6	178.2 ± 6.7 †	176.2 ± 5.7 †
Mass (kg)	79.3 ± 12.7	77.6 ± 11.0	66.9 ± 7.7	65.5 ± 7.2	80.4 ± 12.5 †	78.7 ± 10.6 †
BMI (kg·m^−2^)	25.1 ± 3.1	25.1 ± 2.8	23.1 ± 1.8	23.0 ± 1.1	25.2 ± 3.2 †	25.3 ± 2.8 †
Experience (yr)	8.2 ± 5.4	11.1 ± 5.4 *	6.4 ± 5.5	10.5 ± 4.9	8.3 ± 5.4	11.2 ± 5.4 *
PE mass (kg)	11.5 ± 4.5	11.5 ± 5.0	8.9 ± 3.0	11.8 ± 3.8	11.7 ± 4.6	11.5 ± 5.1

NoCP: No chronic pain; CP: chronic pain; BMI: body mass index; PE: personal equipment. *, differences between NoCP and CP (*p* < 0.05). †, differences between females within same group (*p* < 0.05).

**Table 2 jcm-11-00989-t002:** Cumulative frequency (percentage) of chronic pain according to different grouping variables.

		NoCP	CP1	CP + 1	*p*-Value
Total (*n* = 221)		89 (40.3%)	72 (32.6%)	60 (27.1%)	
Gender	Female (*n* = 18)	7 (38.9%)	8 (44.4%)	3 (16.7%)	0.933
	Male (*n* = 203)	82 (40.4%)	64 (31.5%)	57 (28.1%)	
Job position	WFF (*n* = 181)	75 (41.4%)	54 (29.8%)	52 (28.7%)	0.440
	Foreperson (*n* = 30)	10 (33.3%)	14 (46.7%)	6 (20.0%)	
	Crew leader (*n* = 10)	4 (40.0%)	4 (40.0%)	2 (20.0%)	
Age	18–35 years (*n* = 107)	59 (55.1%) *	24 (22.4%) †	24 (22.4%)	0.000
	36–45 years (*n* = 90)	24 (26.7%) †	36 (40.0%) *	30 (33.3%)	
	>45 years (*n*= 24)	6 (25.0%) †	12 (50.0%) *	6 (25.0%)	
Physical activity	Low (*n* = 49)	23 (46.7%)	13 (26.5%)	13 (26.5%)	0.508
	Moderate (*n* = 128)	49 (38.3%)	47 (36.7%)	32 (25.0%)	
	High (*n* = 44)	17 (38.6%)	12 (27.3%)	15 (34.1%)	
Preventive training	Yes (*n* = 159)	60 (37.7%)	55 (34.6%)	44 (27.7%)	0.435
	No (*n* = 62)	29 (46.8%)	17 (27.4%)	16 (25.8%)	
Experience	≤10 years (*n* = 104)	52 (50.0%) *	28 (26.9%) †	24 (23.0%)	0.021
	>10 yr (*n* = 117)	37 (31.6%) †	44 (36.7%) *	36 (30.8%)	

NoCP: No chronic pain; CP1: chronic pain in one body region; CP + 1: chronic pain in more than one body region. *, higher count in this category than that expected and †, lower count in this category than that expected.

**Table 3 jcm-11-00989-t003:** Cumulative frequency (percentage) of body regions with chronic pain according to grouping variables.

		Lower Limbs	Upper Limbs	Back
Total (*n* = 223)		79 (35.4%)	74 (33.2%)	70 (31.4%)
Gender	Female (*n* = 20)	10 (50.0%) ‡	7 (35.0%)	3 (15.0%)
	Male (*n* = 203)	69 (34.0%)	67 (33.0%)	67 (33.0%)
Job position	Specialist (*n* = 175)	58 (33.1%) †	59 (33.7%)	58 (33.1%)
	Foreperson (*n* = 39)	16 (41.0%) *	13 (33.3%)	10 (25.6%)
	Crew leader (*n* = 9)	5 (55.5%)	2 (22.2%)	2 (22.2%)
Age	18–35 years (*n* = 74)	29 (39.2%) †	23 (31.1%) †	22 (29.7%) †
	36–45 years (*n* = 117)	40 (34.2%) *	38 (32.5%)	39 (33.3%) *
	>45 years (*n* = 32)	10 (31.2%)	13 (40.6%) *	9 (28.1%)
Physical activity	Low (*n* = 46)	15 (32.6%)	14 (30.4%)	17 (37.0%)
	Moderate (*n* = 135)	49 (36.3%)	46 (34.1%)	40 (29.6%)
	High (*n* = 42)	15 (35.7%)	14 (33.3%)	13 (31.0%)
Preventive training	Yes (*n* = 168)	58 (38.5%)	58 (38.5%)	52 (31.0%)
	No (*n* = 55)	21 (38.2%)	16 (29.1%)	18 (32.7%)
Experience	≤10 years (*n* = 81)	29 (35.8%)	26 (32.1%) †	26 (32.1%)
	>10 years (*n* = 142)	50 (35.2%)	48 (33.8%) *	44 (40.0%)

Grouping together of body regions with chronic pain: (i) lower limb (LL) includes hip, knee and ankle; (ii) upper limb (UL) includes the neck, shoulder, elbow, hand and wrist; and (iii) back (B) includes lower back and thoracic spine. *, higher count in this category than that expected (*p* < 0.05). †, lower count in this category than that expected (*p* < 0.05). ‡, differences with back within female group (*p* = 0.007).

## Data Availability

The data presented in this study are available on request from the corresponding author. The data are not publicly available due to privacy restrictions.

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
