# Peer review of "Chronic Pain in Spanish Wildland Firefighters"

_jcm, 2022, doi:10.3390/jcm11040989_

Round 1
Reviewer 1 Report
Dear Authors,
in my opinion, the topic is interesting and the manuscript focused on a specific population not deeply studied. However, besides the intrinsic limitations of retrospective studies, I have several concerns regarding the whole methodological implant of this study, and several critical issues should be addressed to improve the paper.
Major
- This Section should be improved, focusing on the chronic pain consequences in terms of physical function and quality of life, highlighting the role of rehabilitation interventions aiming at preventing and managing this detrimental condition. According to this, you should cite the following references:
- De Sire A et al. Percutaneous Electrical Nerve Stimulation (PENS) as a Rehabilitation Approach for Reducing Mixed Chronic Pain in Patients with Musculoskeletal Disorders. Applied Sciences 2021;11:4257.
- Bendelin N et al. Internet-Delivered Acceptance and Commitment Therapy Added to Multimodal Pain Rehabilitation: A Cluster Randomized Controlled Trial. J Clin Med. 2021;10(24):5872. Published 2021 Dec 14. doi:10.3390/jcm10245872
- Brain K et al. Diet and Chronic Non-Cancer Pain: The State of the Art and Future Directions. J Clin Med. 2021;10(21):5203. Published 2021 Nov 8. doi:10.3390/jcm10215203
- Bernetti A et al. Neuropathic Pain and Rehabilitation: A Systematic Review of International Guidelines. Diagnostics (Basel). 2021;11(1):74. Published 2021 Jan 5. doi:10.3390/diagnostics11010074
- METHODS: please provide the study's ethical approval IRB.
- It might be interesting to provide the sample size calculation concerning the total number of firefighters present in Spain. What response rate did you expect to consider the study viable? Moreover, potential selection bias due to computer skills should be discussed. In particular, in order to fulfill an online questionnaire, more elderly subjects might have been excluded and the sample might be not representative of the population assessed.
- it is not clear how the survey was carried out, concerning surveys already present in the literature? with the help of experts? no previous pilot experience was made to verify face validity, ambiguity of terms, understanding of terms, etc.
- The precise definition of chronic pain should be reported. were the study participants instructed to precisely define chronic pain? Moreover, chronic pain characteristics have not been assessed (i.e neuropathic pain). Self-reported validated scales to characterize chronic pain might significantly improve the study results. I suggest discussing this limitation in the Limitations Subsection.
- the etiological cause of chronic pain has not been assessed (such as traumatic injuries, inflammatory diseases, or other injuries potentially unrelated to work). I suggest discussing this limitation in the Limitations Subsection.
Minor:
- Page 1 Lines 18, 21, 24. Please replace WWFs with the correct acronym WFFs.
- Page 2 Line 47, 48. Please add ‘the’ before ‘lumbar region’ and ‘shoulders’ or remove it before ‘knees’ to be consistent.
- Page 2 Line 52. Please add ‘the’ before ‘upper limbs’, ‘neck’ and ‘back’ or remove it before ‘lower limbs’ to be consistent.
- Page 2 Line 62. Please replace WWFs with the correct acronym WFFs.
- Page 2 Lines 78, 84. Please replace WWFs with the correct acronym WFFs.
- Please provide a reference for the definition of low, moderate, and high physical activity levels.
- Page 3 Line 124. Please replace WWFs with the correct acronym WFFs.
- TABLE 2. Please change the term “Physical training” with “Physical Activity” to be consistent with the “Methods” Section.
- TABLE 2. Please add the p-value in table 2 so that it is handier to read it instead of searching it in the results section.
- INTRODUCTION AND DISCUSSION. There are some sentences not supported by references. Please provide appropriate citations.
Reviewer 2 Report
The topic is of scientific interest and the outcomes may find potential practical application in occupational medicine since they highlight the high prevalence of CP among WFFs.
Specific Comments
- Page 2, line 49: Correct “injures” to “injuries” as well as in page 6, line 207 and page 7, line 251
- Try to state an hypothesis at the end of introduction.
- Methods: It is of great importance to clarify how did you define the “chronic pain” that was assessed in your questionnaires. Moreover, what did you consider as physical activity?
- The size of the sample is rather small as correctly noted in the limitations section
- Do you have any info about the usual training characters of the subjects who were physically active (days per week, type of exercises, intensity etc)?
- Was the reported chronic pain diagnosed by a physician or just self-reported? If it was self-reported, you should mention it in the limitations of the study.
- Data analysis: were there any prior statistical power analyses for the sample of the study?
- Page 3, line 126: change “was” to “were”
- How do you conclude that your findings might provide relevant information to assist in the design of specific training programmes to prevent and improve CP in WFFs since there was no control group of WFFs who did not undertake any physical training programme as part of their working routine or preventive training? Although you discussed it, I think you should not include it in your conclusions.
- Add the limited existing literature in this area to the strengths of your study in the discussion section after the presentation of the limitations.
Round 2
Reviewer 1 Report
Dear Authors,
In my opinion, the topic is interesting and the manuscript is well written.
The results are intriguing and might significantly improve knowledge in this field.
You have significantly improved the manuscript following my suggestions.
Best regards
Reviewer 2 Report
Thank you for your point to point reply and the appled amendments in the manuscript